# Changes in ocean health in British Columbia from 2001 to 2016

**Casey C. O'Hara** [1]*, **Courtney Scarborough**[2], **Karen L. Hunter**[3], **Jamie C. Afflerbach**[2], **Karin Bodtker**[4], **Melanie Frazier**[2], **Julia S. Stewart Lowndes**[2], **R. Ian Perry**[3], **Benjamin S. Halpern**[1,2]

**1** Bren School of Environmental Science and Management, University of California Santa Barbara, Santa Barbara, California, United States of America, **2** National Center for Ecological Analysis and Synthesis, University of California Santa Barbara, Santa Barbara, California, United States of America, **3** Pacific Biological Station, Fisheries and Oceans Canada, Nanaimo, British Columbia, Canada, **4** MapSea Consulting, Vancouver, British Columbia, Canada

* cohara@bren.ucsb.edu

## Abstract

Effective management of marine systems requires quantitative tools that can assess the state of the marine social-ecological system and are responsive to management actions and pressures. We applied the Ocean Health Index (OHI) framework to retrospectively assess ocean health in British Columbia annually from 2001 to 2016 for eight goals that represent the values of British Columbia's coastal communities. We found overall ocean health improved over the study period, from 75 (out of 100) in 2001 to 83 in 2016, with scores for inhabited regions ranging from 68 (North Coast, 2002) to 87 (West Vancouver Island, 2011). Highest-scoring goals were Tourism & Recreation (average 94 over the period) and Habitat Services (100); lowest-scoring goals were Sense of Place (61) and Food Provision (64). Significant increases in scores over the time period occurred for Food Provision (+1.7 per year), Sense of Place (+1.4 per year), and Coastal Livelihoods (+0.6 per year), while Habitat Services (-0.01 per year) and Biodiversity (-0.09 per year) showed modest but statistically significant declines. From the results of our time-series analysis, we used the OHI framework to evaluate impacts of a range of management actions. Despite challenges in data availability, we found evidence for the ability of management to reduce pressures on several goals, suggesting the potential of OHI as a tool for assessing the effectiveness of marine resource management to improve ocean health. Our OHI assessment provides an important comprehensive evaluation of ocean health in British Columbia, and our open and transparent process highlights opportunities for improving accessibility of social and ecological data to inform future assessment and management of ocean health.

## Introduction

Coastal communities depend on oceans to provide a range of ecological, economic, and cultural benefits, including food provision, recreation, jobs and livelihoods, and a sense of identity and well-being. Growing coastal populations and uses of the ocean have led to increasing

**Data Availability Statement:** Original datasets underlying the results presented in the study are available from the original owners (see references).

All code and resulting data generated for our analyses are available at github.com/ohi-science/ohibc.

**Funding:** The authors were supported in this project with funding from the Gordon and Betty Moore Foundation, grant #3538.01 awarded to BSH. The funders had no role in study design, data collection and analysis, decision to publish, or preparation of the manuscript.

**Competing interests:** The authors have declared that no competing interests exist.

impacts from fishing, pollution, climate change and a host of other stressors that threaten the sustainable provision of marine resources and the communities that depend on them [1,2]. Effective marine conservation must balance protection of marine ecosystems with the social and economic needs of coastal communities, informed by reliable and repeated assessment of ecological, economic, and social impacts at temporal and spatial scales relevant to policy-makers [3]. Here we apply the Ocean Health Index (OHI) framework [4,5] over a 15-year period to examine the effects of marine conservation policy, particularly marine protected areas (MPAs) and management of fisheries and aquaculture, on the ocean health of British Columbia (BC), Canada (OHIBC).

The OHI is a holistic framework for assessing and understanding ocean health, defining a healthy ocean as one that "sustainably delivers a range of benefits to people now and in the future" [4]. OHI integrates measures of current status of a wide range of goals and objectives for healthy oceans, along with pressures to those goals and resilience measures designed to improve their status [4]. OHI's increasingly wide application at global [4,5] (see also www.ohi-science.org/ohi-global), regional [6,7], national [8–10], and subnational [11] levels suggests a broadly recognized need for an inclusive, quantitative, and replicable tool for assessing the state of our oceans [12]. However, aside from the global assessment that has been repeated annually since 2012, OHI assessments have been single snapshots in time, focusing only on spatial variation among regions within the study area. These baseline assessments are important, but a key utility of OHI is to track ocean health over time and identify potential drivers, consequences, and tradeoffs.

British Columbia comprises the entirety of Canada's Pacific Ocean coastline and is renowned for forested fjords, iconic marine mammals, and productive fisheries. More than 75% of the province's five million residents live within 50 km of the coast. For millennia, First Nations have relied upon marine ecosystems and resources for food, social, and ceremonial benefits, and these resources continue to play a critical role in communities throughout BC [13]. Extractive activities including fishing and forestry remain vital to BC's economy but impose pressures on the marine ecosystems and compete with recreational, cultural, and spiritual ocean uses [14–16]. Recognition of the potential impacts of these activities on ocean health over time has resulted in increasing attention to protecting and managing BC's ocean resources.

Here the OHI framework is tailored to BC's marine ecosystems and coastal communities (OHIBC) to retrospectively calculate annual ocean health scores over a 15-year period (2001–2016). In collaboration with local experts, we developed goal models to represent BC-specific values and concerns and identified relevant datasets and management targets to quantify the health of the system over the study period. We then compared the results to resilience metrics that account for social well-being, ecological integrity, and regulatory policy to examine the sensitivity of the OHI to marine management actions.

## Methods

The OHIBC assessment comprises eight goals (Table 1) that represent a healthy ocean's ability to sustainably provide a suite of benefits to BC's coastal communities. By focusing on the interconnection of ecological, social, and economic benefits to people, OHIBC (and OHI generally) explicitly recognizes the role of people as an inextricable component of the ocean ecosystem. Here we summarize the methods used to define these goals and calculate this OHIBC assessment; detailed methods and descriptions of data sources can be found in the Supporting Information (SI). As with all OHI assessments, we make all data preparation and analysis code and results freely available with open source software [12]. All of our analyses are coded in the R

**Table 1. Goals and reference targets.**

| Goal | Subgoal | Reference Target |
|---|---|---|
| Habitat Services | Carbon Storage | Coastal forest and salt marsh carbon storage potential greater than or equal to carbon storage potential in 1990 |
| | Coastal Protection | Coastal forest and salt marsh coastal protection potential greater than or equal to coastal protection potential in 1990 |
| Food Provision | Wild-capture fisheries | All fisheries at B/BMSY of 1; all fisheries fished according to DFO harvest control rule; all stocks assessed against defined reference status |
| | Aquaculture | Finfish and shellfish harvests meet potential harvest based on tenure area and growth potential index (mean—1 standard deviation) |
| | Wild-capture Salmon | All indicator stocks at catch targets |
| First Nations Resource Access Opportunities | | Herring spawn index: three-year running mean index at reference index value, calculated as mean index from 1940–1960; Shellfish closures: zero area-weighted average closure days due to sanitation, chemicals, or biotoxins; Salmon: all indicator stocks at or above escapement targets; First Nations commercial fishing licenses: proportion of licenses greater than or equal to proportion of First Nations population in region, with a floor of 15% |
| Coastal Livelihoods | Non-First Nations Livelihoods | For all non-First Nations census subdistricts, total employment rate and median inflation-adjusted wages at or above local mean of prior five year period. |
| | First Nations Livelihoods | For all First Nations census subdistricts, total employment rate and median inflation-adjusted wages at or above local mean of prior five year period. |
| Tourism and Recreation | | Visitations to coastal parks and visitor centers (within 15 km of coast) at or above local mean of prior five year period. |
| Sense of Place | Lasting Special Places | 30% of both coastal marine waters (to 3 nmi offshore) and terrestrial coastal zones (inland coastal watersheds) designated as parks or protected areas (incl. national, provincial, and tribal parks) |
| | Iconic Species | All iconic species extinction threat status at Least Concern |
| Biodiversity | Species | All species extinction threat status at Least Concern |
| | Habitats | Salt marsh extents at 1990 levels; zero trawl pressure on soft bottom habitat and ecologically/biologically significant areas (sponge reefs, deepwater corals, seamounts, hydrothermal vents) |
| Clean Waters | | Nutrient, chemical, pathogen, and marine plastics pollution at zero |

programming language [17]. Our code, along with extensive documentation and intermediate data products, is available on a version-controlled GitHub repository [18]. By making our code and data freely available we hope to encourage others to continue OHIBC assessments into the future; additionally, as better data and knowledge become available, past years can easily be recalculated, as is done with OHI global assessments [5].

## Selection of study regions

The Canadian Pacific exclusive economic zone (EEZ) extends 200 nautical miles from the British Columbia coastline, and is bounded on the north and south by the EEZ of the United States. Within this EEZ, we identified seven assessment regions based upon a combination of conservation planning boundaries, marine ecoregions [19], and biogeographic classifications [20] (Fig 1). We began with four planning regions identified by the Marine Plan Partnership

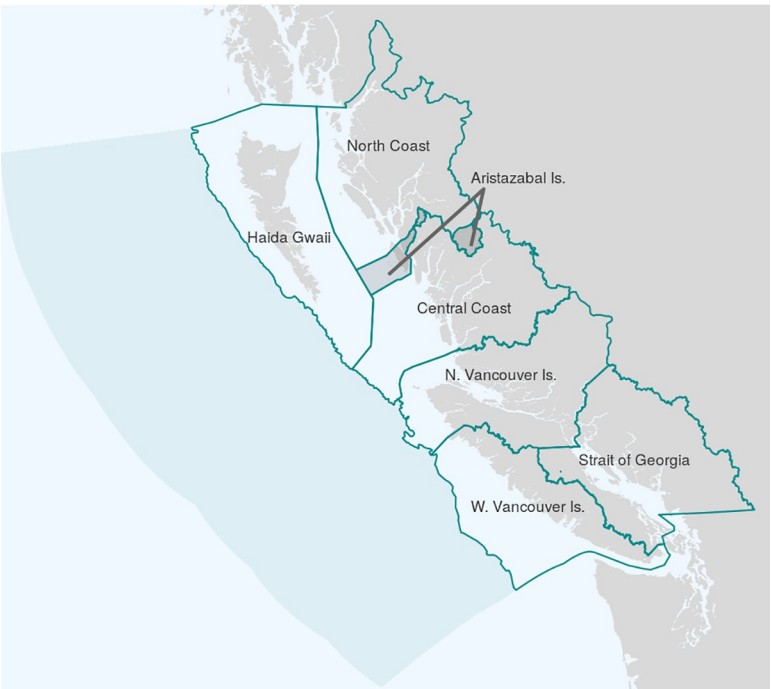

**Fig 1. Assessment regions.** The named regions represent the seven OHIBC regions included in this assessment. The shaded disjoint region, Aristazabal Island, represents an overlap between the MaPP North Coast and Central Coast regions, assessed separately to avoid ambiguity. The shaded offshore region represents the remainder of the Canada Pacific EEZ, not included in this assessment.

for the North Pacific Coast (MaPP), a partnership between BC's provincial government and 17 member First Nations that has developed marine use plans for northern coastal BC: North Coast [21], Haida Gwaii [22], Central Coast [23], and North Vancouver Island [24]. We extended the North Vancouver Island region westward to coincide with the edge of the continental shelf and the break between two marine ecoregions: North American Pacific Fijordland region and Oregon, Washington, Vancouver Coast and Shelf region [19].

Two additional regions define the southern half of BC's coastal waters: Strait of Georgia and West Coast Vancouver Island. To define the boundary between these geographically and economically distinct regions, which were not included in the MaPP planning regions, we relied on the marine ecoregion-defined boundary between the Puget Trough/Georgia Basin region and the Oregon, Washington, Vancouver Coast and Shelf region [19]. Finally, to resolve ambiguity around a small sparsely populated region of overlap between the North Coast and Central Coast MaPP regions, we separated it into its own region, Aristazabal Island. The portion of the Canada Pacific EEZ beyond the break of the continental shelf (Fig 1) is not included in this assessment.

## Structure of OHI

The structure of the Ocean Health Index framework is well-documented for global [4,5] (see also www.ohi-science.org/ohi-global), regional [6], national [8–10], and other subnational assessments [11]. This OHIBC assessment adheres to the same general structure, briefly outlined below, with goals and reference points tailored to BC-specific interests, values, and available data, as expected and encouraged for any regional OHI assessment [7].

The present status $x$ of each goal $g$ is assessed for each of the eight regions in the study area and scored from 0 (poor) to 100 (excellent), as the ratio of the current delivery of benefits $X_{rgn}^g$ relative to a defined reference point $X_{rgn,Ref}^g$ based on the maximum sustainable delivery of those benefits:

$$x_{rgn}^g = \frac{X_{rgn}^g}{X_{rgn,Ref}^g}$$

In addition, each goal receives a score for "likely future status", $\hat{x}_{rgn,F}^g$, estimating delivery of benefits five years into the future (also scored 0 to 100) based upon projected changes in status due to recent trend $T$ (proportional change in status over the previous five years), anthropogenic pressures $p$ that are likely to decrease delivery of benefits, and resilience measures $r$ that are likely to mitigate pressures.

$$\hat{x}_{rgn,F}^g = \frac{1}{(1+\delta)}\left(1 + \beta T_{rgn}^g + (1-\beta)(r_{rgn}^g - p_{rgn}^g)\right)x_{rgn}^g$$

We assume recent trend to be a stronger predictor of likely future status, therefore the relative importance of trend with respect to pressures and resilience, $\beta$, is set to 0.67 (implying that the predictive power of trend is twice that of pressures and resilience) [4]. A discounting term $\delta$ is included but set to zero [4], reflecting equal value placed on future status and current status.

Within each region, each goal receives a total score $I_{rgn}^g$ based upon the average of its current status $x_{rgn}^g$ and its likely future status $\hat{x}_{rgn,F}^g$:

$$I_{rgn}^g = \frac{x_{rgn}^g + \hat{x}_{rgn,F}^g}{2}$$

For each region, an Index score is calculated as the unweighted average (all weighting factors $\omega_g$ set to 1) of all $N$ goal scores assessed within the region:

$$I_{rgn}^{Index} = \frac{1}{\sum_{g=1}^{N}\omega_g}\sum_{g=1}^{N}\omega_g I_{rgn}^g$$

Goal scores for the entire OHIBC study area (including overall Index score $I_{BC}^{Index}$) are calculated as the mean of regional goal scores weighted by the marine area $A_{rgn}$ of the $R = 7$ coastal regions:

$$I_{BC}^g = \frac{1}{\sum_{rgn=1}^{R}A_{rgn}}\sum_{rgn=1}^{R}A_{rgn}I_{rgn}^g$$

## Identifying datasets for spatial-temporal assessment

For OHIBC, our primary objectives were to examine changes in ocean health over time and identify possible interactions between ocean health status, human and natural pressures on the marine system, and major changes in ocean policy and management. We accomplished this by building a set of goal-specific indicators that were used to calculate a score for each goal. We sought data that were updated annually to best represent how an indicator was changing through time. While annual temporal extent and resolution were priorities in the data selection process, we also had to balance these against our criteria for sufficient spatial extent and resolution to allow region-level assessment and inter-region comparisons. While many available datasets met either our spatial or temporal criteria, not all data were able to meet both. In

such cases, datasets were chosen based on careful consideration of the tradeoffs among spatial extent and resolution, temporal extent and resolution, and how well the data represented the needs of the assessment (see SI methods, S13 and S14 Tables).

Raw data were prepared into "data layers" to be used as inputs to OHIBC calculations, via a combination of cleaning, error checking, transforming, rescaling, and/or gapfilling. Each data layer contains information used to calculate OHI scores, and typically includes values corresponding to specific OHIBC regions and years (Fig 2). S1—S3 Tables list the full set of data layers for status, pressures, and resilience respectively.

OHI goal models require a reference point or target that is used to rescale scores to range from 0 to 100. Reference point selection is driven by the quality of available data, the needs of

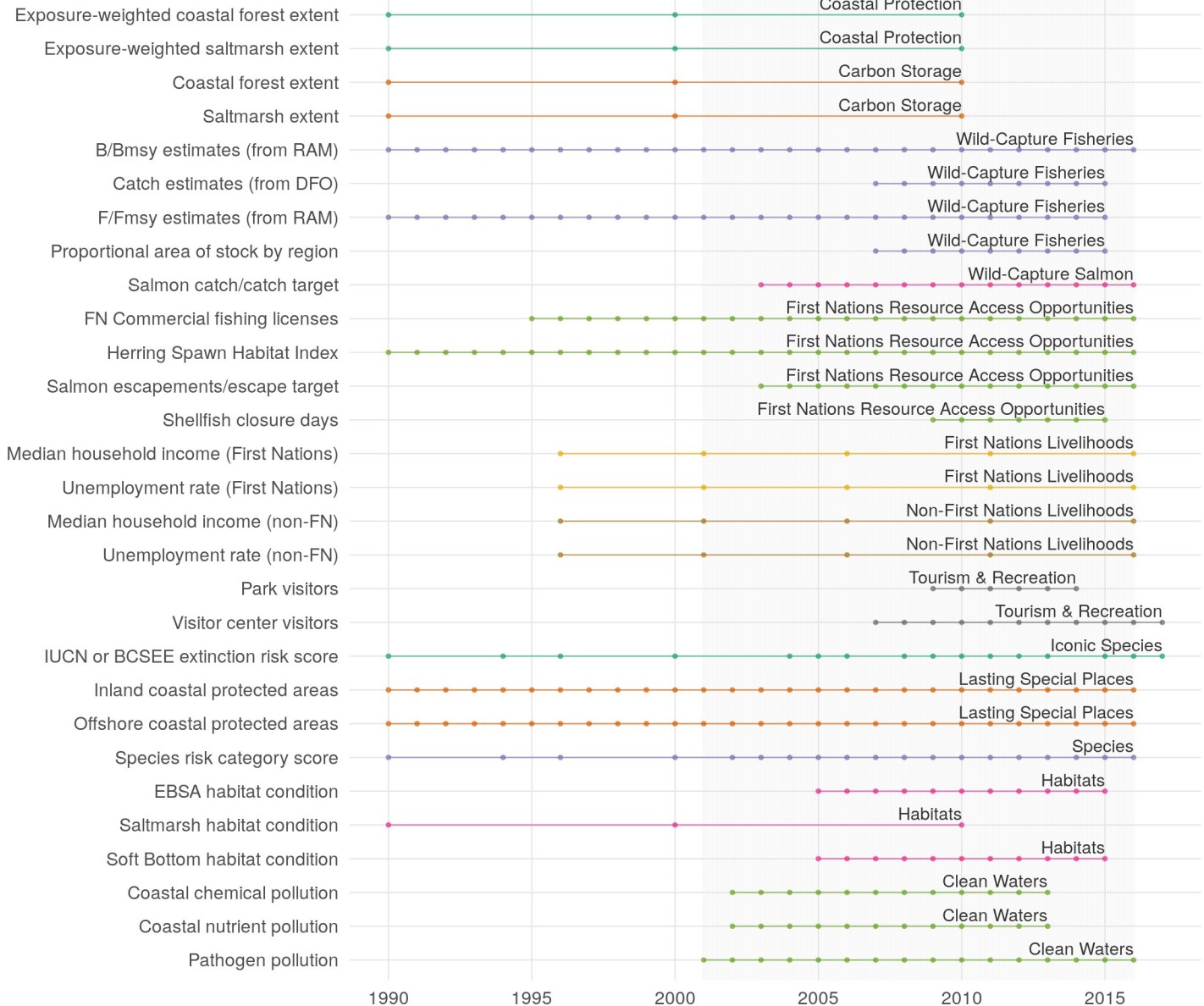

**Fig 2. Data layers used to calculate OHIBC goals and subgoals.** Available data years are indicated as points. Some datasets begin before 1990 but these points are omitted for clarity. Color and text identify goals. Shaded region indicates study period, while data prior to 2001 were used to inform historic baselines and recent trends. Static data layers (e.g., region area km$^2$) are not shown.

the goal model, and when known, social or ecological thresholds. Changing a reference point can have significant impact on a goal's score [25], therefore transparency about the selection of a reference point is critical. Table 1 qualitatively describes the reference points used to score each OHIBC goal; detailed descriptions for reference points and methods for each data layer can be found in the SI and in the code [18].

Wherever possible, data for a given year were used to assess the scores for that year. Thus, OHIBC scores for each year were based on data on fisheries, wages, habitat coverage, etc. from the same year. In many cases, time-series data were periodic (e.g., Canada census data taken every five years) or truncated (e.g., data series began after 2001 or ended before 2016) (Fig 2). In some cases, data were both periodic and truncated (e.g., 30m gridded land use rasters for 1990, 2000, and 2010). To address incomplete data sets, we applied two gapfilling procedures chosen for simplicity and transparency. For periodic data, we estimated intervening years using a linear interpolation between available data years. For truncated data, we typically expanded the time series using last observation carried forward and/or next observation carried back extrapolation. The SI Methods detail specific data preparation methods for each data layer and goal model, including where and how these gapfilling techniques were applied.

## OHIBC goals

The OHI framework is designed to be flexible and tailorable to account for local goals and values for healthy oceans and to better leverage the best available local-scale data [7,8]. To incorporate values across a broad range of people who benefit from a healthy marine social-ecological system, we engaged several user communities to select and design OHIBC goals. In partnership with MaPP representatives, we performed a crosswalk of the ten original OHI goals against the objectives stated within the four regional MaPP plans [21–24] to identify opportunities for adapting goals to local values. Additionally, we crosswalked OHI goals with the themes of the Coastal Ocean Research Institute (CORI) OceanWatch project, including calculation of selected goals to support the narratives presented in the OceanWatch Howe Sound project [26]. The final set of eight goals for OHIBC (Table 1) closely match the ten goals from the global OHI [4], with three important differences.

First, we combined the Carbon Storage and Coastal Protection goals into a single Habitat Services goal. This decision was made in consultation with partners and stakeholders in the region and was meant to reflect the local conceptualization of salt marsh and coastal forest habitats providing a suite of vital services that should be considered comprehensively as a single goal. Next, we excluded the Natural Products goal because it was not relevant to this region, reducing the number of goals to eight. Finally, we modified two goals to better reflect the priorities of the region, one by adding an additional subgoal in the Food Provision goal focused on wild-caught salmon to acknowledge the importance of these iconic wild fish to BC seafood provision, and the other by refocusing the global Artisanal Fishing Opportunities goal on First Nations Resource Access Opportunities. Detailed methods and data sources for each goal and subgoal are included in the SI, with each briefly summarized below.

The **Habitat Services** goal contains two subgoals, **Carbon Storage** and **Coastal Protection**, that measure the ability of marine-associated biogenic habitats to sequester carbon and protect coastlines from erosion and storm surge, respectively. Both subgoals were calculated using change in salt marsh and coastal forest habitats, weighted by either carbon sequestration potential or coastal protection value accordingly. Ideally, condition of seagrasses and other key habitats (e.g., changes in extent or time series of significant pressures) would be included in this goal, as in other OHI assessments [4]; however, sufficient data to inform understanding of condition over time were not available.

The **Food Provision** goal comprises three subgoals—Wild-Capture Fisheries, Aquaculture, and Wild-Capture Salmon—that together measure the realized harvest from these sectors against the maximum sustainable provision of seafood for both domestic consumption and export. Scores for the three subgoals are combined in an unweighted average to determine the Food Provision score.

The **Wild-Capture Fisheries** subgoal relies on stock assessments and landings data to score each region's fishing of harvested stocks relative to Maximum Sustainable Yield (MSY), based upon biological ($B/B_{MSY}$) and fisheries management ($F/F_{MSY}$) targets, where $B/B_{MSY}$ is the current biomass ($B$) relative to the amount that can be sustainably harvested ($B_{MSY}$), and $F/F_{MSY}$ is the current fishing mortality rate ($F$) compared to the optimal rate at MSY ($F_{MSY}$). Spatially explicit landings data by region were provided by Fisheries and Oceans Canada (DFO) for a subset of important commercial species (see SI methods); $B/B_{MSY}$ and $F/F_{MSY}$ estimates, available for a subset of harvested stocks, were obtained from the RAM Legacy Stock Assessment database [27]. A large number of BC fish stocks lack quantitative stock assessments relative to an MSY reference point; therefore, we include an "unassessed stock" penalty to account for the uncertainty and risk inherent in exploiting stocks without data-informed targets. This penalty has not been used in previous OHI assessments. Stock assessment metrics $B/B_{MSY}$ and $F/F_{MSY}$ are used to assign a score for each stock between 0 and 1 based on proximity to a harvest control rule, with scores designed to penalize both overexploitation and underexploitation (S2 Fig). A region's Wild-Capture Fisheries status is calculated as the average stock score, weighted by each stock's contribution to the total catch for the region, and then multiplied by the unassessed stock penalty. Several commercially and culturally important stocks with non-MSY-based assessment methods (i.e., Pacific herring, geoduck, Dungeness crab, shrimp species) were unable to be scored but neither are they penalized.

The **Aquaculture** subgoal compares the production of bivalve and finfish marine aquaculture to a reference point based on each region's mean production potential, calculated using a spatially explicit global model of biological production potential [28], and size and location of existing aquaculture tenures. Regions with no active aquaculture tenures were not assigned a score.

The **Wild-Capture Salmon** subgoal considers salmon separately from other fisheries because management of wild salmon largely relies on estimating the number of returning salmon to their natal rivers, often from multiple stocks that are homing to vastly different regions or areas within a region, while fisheries for these stocks are occurring. As a sustainability measure, stocks and stock complexes are monitored 'in-season' and catches are adjusted to account for the highly variable returns observed for many BC salmon stocks. For a small number of fisheries, catch targets have been developed through modeling of stock complexes or by assigning an operational target based on homing dynamics and location. A set of indicator salmon fisheries are scored based on catch estimates and target information, and these indicator scores are averaged across the entire study region as a proxy for sustainability of harvest of major wild salmon fisheries. Accounting for wild-capture salmon as a separate sub-goal allows transparency in the inclusion of this information into the OHIBC assessment while also communicating the iconic importance of salmon fisheries to BC.

A modified goal, **First Nations Resource Access Opportunities**, recognizes that for millennia First Nations have depended on marine resources for non-economic food, social, and ceremonial purposes [13]. In addition to food security and sovereignty [29], inclusive marine resource access broadly supports socioeconomic values, cultural values, and social capital [30], acquisition and transfer of traditional ecological knowledge [31], and political representation in decision making [32]. First Nations Resource Access Opportunities assesses access to four marine resources that continue to be highly valued by BC First Nations [21–24], regardless of

realized exploitation of these resources: availability and access to herring roe on kelp (based on herring spawn index); availability and access to healthy salmon runs (based on salmon escapement estimates and escapement targets for a set of indicator stocks, with escapement being the number of salmon estimated to reach their spawning grounds in a year); spatial access to unpolluted shellfish harvesting areas (based on shellfish closures related to contamination and biotoxins); and resource and spatial access to the livelihoods provided by commercial fisheries (based on commercial fishing licenses issued for First Nations by DFO). The four components are scored separately and averaged to provide an overall score.

The **Livelihoods** goal incorporates employment rate and inflation-adjusted wage data for coastal communities to measure the availability of high-quality jobs. This goal is calculated as two separate subgoals, **First Nations Livelihoods** (focused on census subdivisions specifically classified as First Nations found within regional boundaries) and **Non-First Nations Livelihoods**. While not all jobs in coastal communities are directly tied to the ocean, we assume that the overall economic vibrancy is dependent on marine sector opportunities; however, this assumption is unlikely to be accurate in large urban centers, e.g., Vancouver. Ideally, we would also include non-monetary benefits in these calculations to represent the diversity of ways the ocean provides livelihoods to people across these regions [13]; however, we were unable to identify quantitative data to communicate this type of information. In global assessments, coastal economies are also assessed as part of the Livelihoods and Economies goal; here we were unable to assess coastal economies because marine sector specific revenue data were not available at sufficient spatial or temporal resolution.

While tourism and recreation can provide revenue and jobs for coastal communities, the **Tourism & Recreation** goal focuses on the value that people place on coastal experiences, measured by the number of visitors entering BC's coastal parks and visitor centers each year. Scores are based on "no net loss" relative to the average visitation over the preceding five year period. Because our dataset begins in 2007, the no net loss reference point automatically scores all regions at 100 for that year. Metrics based on commercial information, such as hotel stays or flight arrivals, may underestimate the value of recreation to local citizens; park and visitor center visitations integrate the preferences of both residents and visitors alike.

The remaining goals were assessed in the same manner as the global assessment but with regional, higher resolution information. **Sense of Place**, which measures the intrinsic value of how a healthy ocean provides a sense of identity and well-being, tracks the extent of protected seascapes and conservation status of iconic marine species. In conjunction with CORI and the Vancouver Aquarium Marine Science Centre, we identified a list of 42 species (and four subpopulations of *Orcinus Orca*) (see SI methods) of distinctive cultural importance to the people of BC; the **Iconic Species** subgoal measures the average conservation status of these species within each region, based upon Committee on the Status of Endangered Wildlife in Canada (COSEWIC) and NatureServe [33] and International Union for Conservation of Nature (IUCN) assessments [34] (prioritizing local COSEWIC scores over global IUCN assessments). The **Lasting Special Places** subgoal calculates the proportion of area designated as park or protected area for natural or cultural values by federal, provincial, or First Nations governance. The proportion of protected area is compared to a 30% target (as done in the global OHI assessment) and calculated separately within both coastal marine areas (to 3 nmi offshore) and coastal watersheds (sub-watersheds intersecting a 1 km buffer inland from the coastline). The Lasting Special Places score is the mean of the coastal marine and coastal watershed scores.

The **Biodiversity** goal assesses the condition of both **Species** and biogenic **Habitats**. The Species subgoal, as in the global assessment, compares area-weighted average conservation status of all assessed species in a region. Conservation status for each species is based upon an assessment from either COSEWIC/NatureServe [33] or IUCN [34] (prioritizing local

COSEWIC scores over global IUCN assessments). The Habitats subgoal, as in the global assessment, compares habitat condition to reference points based on historic extent and lack of human pressures. Habitats assessed for the Habitats sub-goal include salt marsh, coastal forests, seamounts, sponge reefs, deep sea coral, hydrothermal vents, and soft bottom seafloor habitat.

The **Clean Waters** goal assesses the magnitude of pollution in coastal waters according to four components: nutrient pollution from land-based agricultural runoff, chemical pollution from land-based and ocean-based sources, marine plastic debris, and human pathogens from untreated wastewater. Each of these components is scored from 0 to 1, where 1 indicates no contaminant present and 0 indicates an unacceptable level of contaminant (dependent on contaminant; see SI methods for specific reference points). A poor score in any single component is likely to greatly reduce the perceived cleanliness of the water; as such, the total score is a geometric mean of the four component scores, rescaled 0 to 100.

## Pressures

Quantitative estimates of pressures and resilience are used to calculate likely future status. The pressures dimension of OHI estimates the potential negative impacts of a suite of stressors on delivery of each ocean health goal, accounting for the magnitude of each stressor and sensitivity of the goal to that stressor [4]. Following the methods of the global OHI [4,5], pressures accounted for in this OHIBC assessment include pollution (chemical, nutrient, pathogens, and trash), aquatic invasive species, aquaculture impacts, habitat destruction, commercial fishing, climate change (ocean acidification, sea level rise, and extreme anomalies of sea surface temperature and ultraviolet radiation), and social pressures. All pressure data layers are scored for each region from 0 to 1 (with 1 being the highest possible negative impact); pressures layers and data sources are listed in S2 Table, and methods and reference points for pressures are described in detail in the SI Methods.

## Resilience

The resilience dimension of OHI uses three categories of resilience metrics to estimate the ability of the social-ecological system to mitigate stressors and pressures on delivery of ocean-related benefits: ecological integrity (i.e., the ability of an ecosystem to support and maintain ecological processes and a diverse community of organisms [4]), social resilience, and governance (i.e., management actions) [4]. Ecological integrity and social resilience imbue general resilience, essentially acting as buffers to absorb and/or adapt to the entire suite of pressures acting on the ecosystem. Targeted management actions, on the other hand, are designed to directly address the impacts of specific pressures on ocean health goals.

Ecological integrity in OHIBC is based on the relative condition of marine biodiversity. Data layers are calculated in the same manner as the Species subgoal, calculated for both a 3 nmi coastal buffer (to address coast-specific pressures, e.g., intertidal habitat destruction from trampling), and for the entire region (to address wider-ranging pressures, e.g., soft bottom habitat destruction from trawling). Social resilience is based on the Community Well Being Index, a composite index of income, education, housing, and labour force activity to assess "well-being" of Canadian communities [35] at the census subdistrict level. Similar to census data used in First Nations Livelihoods and Non-First Nations Livelihood subgoals, resilience scores were calculated separately for well-being of First Nations communities (to address pressures specifically affecting First Nations-specific goals) and communities overall (to address pressures affecting more general goals).

To quantify governance-based resilience for OHIBC, we sought evidence for 1) existence of targeted policy, regulation, or management action, 2) enforcement of the policy, and 3)

compliance with the policy [4]. Governance-related resilience measures include establishment of MPAs, regulation and enforcement of commercial fishing (based on Groundfish fleet only including fisheries officers, at-sea observer coverage, and a groundfish trawl fishery habitat protection agreement), and regulation and enforcement of marine aquaculture (including health audits and reporting compliance). While governance and ocean management are clearly major determinants of how healthy and sustainable we should expect our oceans to be, data for such metrics are often scarce. When searching for governance data in British Columbia we looked to metrics that most directly targeted the pressures included in the Index. Monitoring and compliance data were available for multiple years, communicating change over time; however, these data were reported at the province level, so spatial variation is not observable, and therefore identical scores were assigned across all regions for each year. As was done for pressures, resilience indicators are rescaled from 0 to 1. All resilience layers and data sources are listed in S3 Table, and resilience methods and reference points are described in detail in the SI Methods.

In British Columbia, ecological integrity and institutional and social organization are at relatively high levels, such that summed resilience scores for a particular goal often exceed total pressures on that goal. However, resilience indicators are intended to directly address, as much as possible, specific pressures [4]; therefore, once a pressure has been adequately addressed, resilience should not further affect the score, to prevent artificial inflation of likely future status. In a significant modification from past OHI methods [4], where total resilience scores were allowed to exceed total pressures scores, we have capped resilience such that ($r$ $-p$)$\leq$0 (i.e. $r\leq p$) when calculating the likely future status for a given goal.

## Examining OHIBC sensitivity to management

By calculating time series status, pressures, and resilience scores for OHIBC goals, we were able to examine whether ocean health metrics changed as one might expect as a result of management actions. We assumed a causal sequence in which 1) a management action is taken at some point in time (resulting in a small but immediate boost to resilience scores regardless of effectiveness); 2) if the management is effective, the pressure addressed by the management action will decrease after some time lag (reflected in a reduced pressure score in some future year); 3) if the decline in pressure is sufficient, indicators of ocean health will improve after an additional interval of time (reflected in an improvement in goal status in some future year). We tested each link in this causal sequence using linear regression. To test for change related to management actions independent of other resilience effects, we separated regulatory resilience (i.e. specific resilience [36]) from ecological and social resilience (i.e. general resilience [36]).

To test the effect of management on ecological pressures (linking steps 1 and 2 in our causal chain), we performed a linear regression of observed proportional change in pressure $\frac{p_{ecol,t+\lambda}}{p_{ecol,t}}$ at time $t+\lambda$ against resilience score components $r_{reg,t}$, $r_{ecol,t}$, and $r_{soc,t}$ at time $t$ for each goal, allowing lag time $\lambda$ to vary from one to six years:

$$\frac{p_{t+\lambda} - p_t}{p_t} = \alpha + \beta_1 r_{reg,t} + \beta_2 r_{ecol,t} + \beta_3 r_{soc,t}$$

Each model/goal/lag combination was subjected to leave-one-out cross validation to inform model selection. Social pressures were excluded due to collinearity with social resilience, as they rely on the same underlying data set. Additionally, our data for ecological resilience is based on species condition, which is essentially constant over the time period; therefore, $r_{ecol,t}$ was omitted from the regression to avoid singularity issues.

To test the effect of changes in pressure on ocean health (i.e., linking steps 2 and 3 in our causal chain), we regressed pressures $p_t$ against observed proportional change in status $\frac{x_{t+\lambda}}{x_t}$ for each goal, again allowing lag time $\lambda$ to vary from one to six years.

$$\frac{x_{t+\lambda} - x_t}{x_t} = \alpha + \beta p_t$$

This regression was calculated separately for ecological pressures $p_t = p_{ecol,t}$ and combined ecological and social pressures $p_t = p_{ecol,t} + p_{soc,t}$.

## Results

OHIBC Index scores for 2016, the most recent year included in the assessment, are generally high, and similar from region to region, scoring from 78 (North Vancouver Island) to 86 (Haida Gwaii), resulting in an overall OHIBC Index score $I_{BC}^{Index}$ of 83 (Fig 3). Scores for all goals across all years and regions are presented in S4 Table.

Across the study period, Index scores for British Columbia $I_{BC}^{Index}$ ranged from 71 in 2002 to 83 in 2016 (Fig 4), significantly increasing by 0.6 points per year ($R^2 = 0.696$) (S5 Table). At the region level, Index scores $I_{rgn}^{Index}$ ranged from 68 (North Coast, 2002) to 87 (West Vancouver Island, 2011) (Fig 4). Mean index scores $\overline{I}_{rgn}^{Index}$ across all years ranged from 76 (Central Coast) to 81 (West Vancouver Island). Index scores in all regions except West Vancouver Island showed significant increasing trends greater than 0.50 points per year over the study period, with Haida Gwaii improving at 0.94 points per year (S6 Table).

BC-wide, scores for individual goals $I_{BC}^{g}$ (Fig 4; see also S1 Fig for subgoal scores) ranged from 32 (Food Provision in 2002) to 100 (Habitat Services, all years); highest-scoring goals, averaging across the study period, were Habitat Services (100) and Tourism & Recreation (94); lowest-scoring goals were Sense of Place (61) and Food Provision (64) (Fig 4). Region goal scores $I_{rgn}^{g}$ ranged from 28 (Food Provision in North Coast, 2002) to 100 (Habitat Services for all years in all regions except Strait of Georgia). All regions assessed for Tourism and Recreation begin at 100 for the first year of data (2007) and often remained high.

Overall scores for Habitat Services showed a small but statistically significant decline of 0.008 points per year; Livelihoods increased by 0.62 points per year; and Sense of Place increased by 1.43 points per year. Other goals did not see any statistically significant change at the BC scale (S5 Table). At the regional scale, goal score trends varied greatly by region and by goal (S6 Table). Notable significant downward trends include a 0.63 point per year drop in Sense of Place for West Vancouver Island, a 0.24 point decline per year in Biodiversity for Central Coast, and a 0.71 point per year decrease in First Nations Resource Access Opportunities in the North Coast. Notable upward trends include Livelihood increases of 0.55, 0.56, 0.61, and 1.58 points per year for Haida Gwaii, West Vancouver Island, North Vancouver Island, and North Coast respectively; and Sense of Place increases of 0.53, 1.63, 2.62, 2.84, and 0.53 points per year for North Vancouver Island, North Coast, Haida Gwaii, and Central Coast, respectively.

### Management impacts

The effects of resilience factors on ecological pressures vary widely by goal. Table 2 shows coefficients for the best supported model and lag value $\lambda$ within each goal, lowest root-mean-square error (RMSE) calculated in the leave-one-out cross validation. (S7 Table shows for each goal the best model, selected among all combinations of the possible resilience components based on minimized RMSE, for each value of $\lambda$). For all goals, RMSE was minimized with a lag of one year. Significant negative coefficients on regulatory resilience (indicating that improved

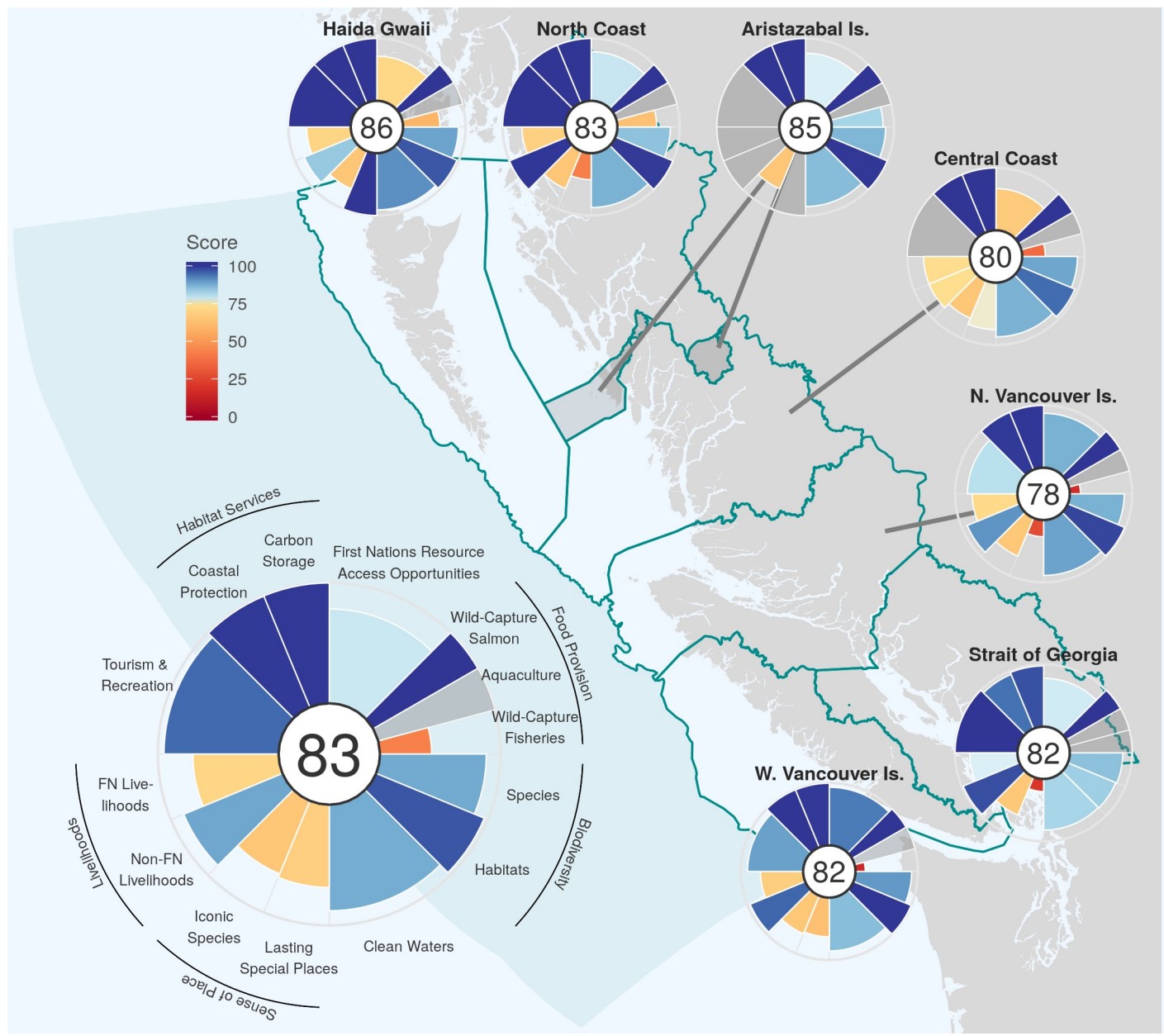

**Fig 3. Spatial patterns in OHIBC scores for 2016.** The large flower plot indicates overall OHIBC score ("Index", center number), with petal length and color indicating relative values (0–100) for each goal and subgoal. Gray goal petals represent goals not calculated for that region. Small flower plots indicate region Index score (center number) and goal values (petal lengths and colors). Goal petals on small plots correspond with goals indicated on the large flower plot. The width of each petal represents the contribution to the Index score.

management correlates with decreasing pressures) were identified for Coastal Protection, Carbon Storage (identical to Coastal Protection as the two Habitat Services goals are subjected to the same pressures and resilience layers), and Wild-Capture Fisheries. Pressures on Coastal Protection and Carbon Storage also showed a significant negative correlation with social resilience. Negative intercepts on Coastal Protection, Carbon Storage, Wild-Capture Fisheries, and Aquaculture indicate additional systematic reductions in pressures separate from resilience components. Models with longer lags (S7 Table) often also showed negative coefficients on resilience components and intercepts at various levels of significance.

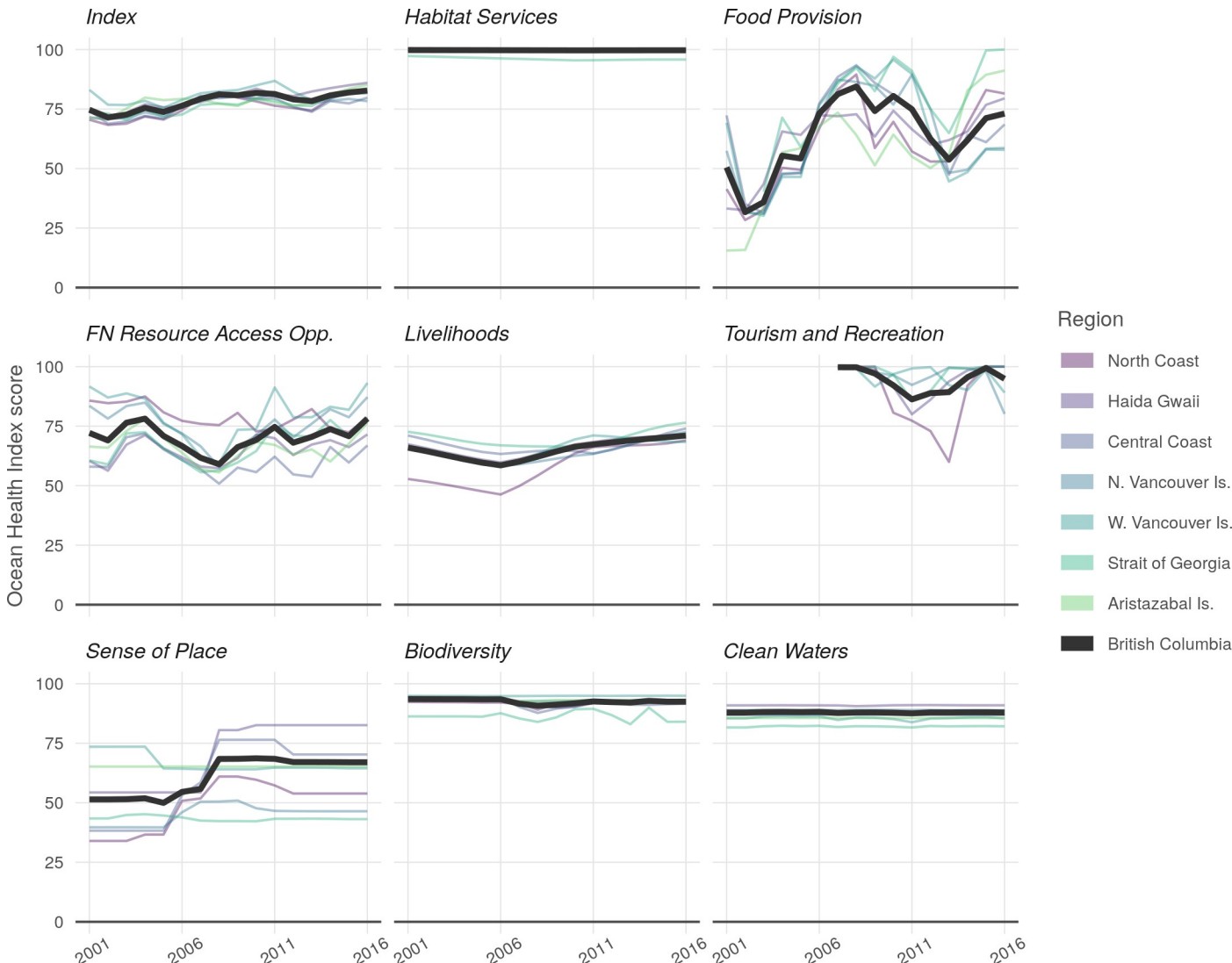

**Fig 4. Index scores and goal scores over time.** The heavy dark line indicates BC-level scores for each goal; the thinner lines represent region-level scores for each goal.

While a reduction in pressures would be expected to improve future status scores, the effects of combined ecological and social pressures on changes in status are generally small and not statistically significant (Table 3), even when focusing on ecological pressures only (i.e., those addressed by regulatory resilience) (Table 4). The Coastal Protection, Carbon Storage, and Clean Waters goals showed statistically significant but positive correlations between ecological pressures and future status (i.e., in the opposite direction from the expected effect), though the small coefficients ($\ll 1\%$) suggest minimal ecological significance. Models with longer lags (S8 Table) show the same basic pattern across goals.

## Discussion

OHIBC scores driven by ecosystem conditions (i.e., Habitat Services, Biodiversity, Clean Waters) were generally high and stable over time, which is consistent with the relatively slow response of these ecological systems to pressures, particularly in a resilient system. Strait of

**Table 2. Proportional change in pressure (at time t + λ) vs. resilience (at time t).**

Fixed effect coefficients on region and year are omitted for clarity.

| goal | subgoal | λ | intercept | reg resil | soc resil | adj.R² |
|---|---|---|---|---|---|---|
| Habitat Services | Coastal Protection | 1.000 | 0.4048* | -0.0795* | -0.7165* | 0.639 |
| | Carbon Storage | 1.000 | 0.4048* | -0.0795* | -0.7165* | 0.639 |
| Food Provision | Wild-Capture Fisheries | 1.000 | -0.0438* | -0.2209* | - | 0.509 |
| | Aquaculture | 1.000 | -0.0600** | - | - | 0.611 |
| | Wild-Capture Salmon | 1.000 | 0.0013 | -0.1237˚ | - | 0.447 |
| First Nations Res. Access Opp. | | 1.000 | -0.2188 | - | 0.2381 | 0.330 |
| Tourism & Recreation | | 1.000 | -0.0227˚ | - | - | 0.496 |
| Sense of Place | Iconic Species | 1.000 | -0.0063 | - | - | 0.218 |
| | Lasting Special Places | 1.000 | -0.0101 | - | - | 0.316 |
| Biodiversity | Species | 1.000 | -0.0783 | - | 0.0779 | 0.491 |
| | Habitats | 1.000 | -0.0277 | - | - | 0.244 |
| Clean Waters | | 1.000 | -0.0014 | 0.0026 | - | 0.124 |
| | | 1.000 | 0.0004 | - | - | 0.124 |

Significance codes

***: p < 0.001

**: p < 0.01

*: p < 0.05

˚: p < 0.1

Georgia and West Coast Vancouver Island regions typically scored lower than other regions on Biodiversity and Clean Waters due to the increased pressures imposed by higher population density and development. Goals related to resource extraction (i.e., Food Provision, First Nations Resource Access Opportunities) varied more strongly from region to region and from

**Table 3. Proportional change in status (at time t + λ) vs. all pressures (at time t).**

Fixed effect coefficients on region and year are omitted for clarity.

| goal | subgoal | λ | intercept | soc+ecol prs | adj.R² |
|---|---|---|---|---|---|
| Habitat Services | Coastal Protection | 1.000 | -0.0002 | | 0.540 |
| | Carbon Storage | 1.000 | -0.0001 | | 0.553 |
| Food Provision | Wild-Capture Fisheries | 1.000 | -0.0699 | | 0.076 |
| | Aquaculture | 2.000 | -0.2589 | | 0.432 |
| | Wild-Capture Salmon | 2.000 | 0.4682*** | 0.0000 | 1.000 |
| First Nations Res. Access Opp. | | 1.000 | 0.0006 | | 0.486 |
| Tourism & Recreation | | 1.000 | -0.4966 | 0.0180 | -0.038 |
| Sense of Place | Lasting Special Places | 1.000 | 0.0792 | | 0.211 |
| Biodiversity | Species | 1.000 | -0.0007*** | | 0.781 |
| | Habitats | 4.000 | -0.0632* | 0.0063** | 0.252 |
| Clean Waters | | 1.000 | -0.0227* | 0.0007* | 0.152 |

Significance codes

***: p < 0.001

**: p < 0.01

*: p < 0.05

˚: p < 0.1

**Table 4. Proportional change in status (at time t + λ) vs. ecological pressures (at time t).**

Fixed effect coefficients on region and year are omitted for clarity.

| goal | subgoal | λ | intercept | ecol prs | adj.R² |
|---|---|---|---|---|---|
| Habitat Services | Coastal Protection | 1.000 | -0.0006˚ | 0.0001** | 0.582 |
| | Carbon Storage | 1.000 | -0.0001˚ | 0.0000* | 0.580 |
| Food Provision | Wild-Capture Fisheries | 1.000 | -0.0699 | | 0.076 |
| | Aquaculture | 2.000 | -0.2589 | | 0.432 |
| | Wild-Capture Salmon | 2.000 | 0.4682*** | 0.0000 | 1.000 |
| First Nations Res. Access Opp. | | 1.000 | 0.0006 | | 0.486 |
| Tourism & Recreation | | 1.000 | 0.0024 | | -0.045 |
| Sense of Place | Lasting Special Places | 1.000 | 1.7810 | -0.0803 | 0.212 |
| Biodiversity | Species | 1.000 | -0.0007*** | | 0.781 |
| | Habitats | 4.000 | 0.0712 | -0.0059 | 0.195 |
| Clean Waters | | 1.000 | -0.0789*** | 0.0025*** | 0.423 |

Significance codes

***: $p < 0.001$

**: $p < 0.01$

*: $p < 0.05$

˚: $p < 0.1$

year to year as these biotic marine resources respond on a time scale of months or years rather than decades, determined by life history traits of species.

Habitat Services (Coastal Protection, Carbon Storage), based on loss of coastal forests and salt marsh relative to a 1990 baseline, score high province-wide due to the low levels of development across the majority of BC's coast, even in the more densely populated Strait of Georgia region. The relatively recent baseline, however, obscures the effects of past development. Historical studies of land use around the Strait of Georgia estimate as much as 30% loss of salt marsh relative to early 20th century levels [37]. As similar reference points were unavailable for other regions, we were unable to compare current conditions to a consistent historic baseline across all regions. Similarly, decades of timber extraction [38] precede the dataset used in our estimates.

Food provision scores vary drastically from region to region (Fig 3) and year to year (Fig 4), with all three components (Wild-Capture Fisheries, Aquaculture, and Wild-Capture Salmon) fluctuating over time (S1 Fig). Prior to 2008 there appears to be an overall rise in scores (7.1 points per year, BC-wide), followed by a decline since (significant declines in Central Coast, North Vancouver Island, and West Vancouver Island of 3.9, 4.8, and 6.2 points per year, respectively). A substantial portion of wild-caught species lack up to date stock assessments [39] and thus are only accounted for in the Wild-Capture Fisheries score via the unassessed stock penalty (see SI methods). Pacific hake plays a major role in the overall abundance of biomass available to Wild-Capture Fisheries for West Vancouver Island and North Vancouver Island regions, though natural variation in stock availability due to temperature-mediated northward migrations limit the amount of hake available to be caught in the Central Coast, North Coast, and Haida Gwaii regions. Note the lack of a Wild-Capture Fisheries score for Strait of Georgia, which holds across all years (Figs 3 and S1); the bulk of Strait of Georgia harvest (>90%, often 100%) comes from stocks that are unassessed or for which no operationally defined fishery target or limit reference points (e.g. MSY) were available.

Southern and central regions (Strait of Georgia, West Vancouver Island, North Vancouver Island, Central Coast) support aquaculture for both finfish, for which production rates are

often close to estimated potential (based on growth potential index [28]), and shellfish, for which production potential is far below modeled potential [28]. Based on tenures data, aquaculture in Haida Gwaii and North Coast regions is limited to shellfish (though production data from 2011–2015 shows no production for North Coast). Special Management Zones noted in the MaPP regional plans [21–24] indicate interest in future development of aquaculture in all MaPP regions.

There appears to be a general decline in First Nations Resource Access Opportunities scores leading up to 2008, followed by an increase since. Prior to 2008, BC-wide scores declined at 2.0 points per year; scores for North Coast, North Vancouver Island, and West Vancouver Island all showed significant declines, up to 4.9 points per year for West Vancouver Island. From 2008 to 2016, BC-wide scores increased at 1.6 points per year; scores for Central Coast, North Vancouver Island, West Vancouver Island, and Strait of Georgia all showed significant increases greater than 1.3 points per year. For all four components, perceptions of access are likely most relevant at a community scale rather than a broader regional scale; but due to data limitations and resolution, all components are scored based on values aggregated to each region, except salmon availability, which is aggregated to the overall OHIBC study region (similar to Wild-Capture Salmon subgoal). For all regions except the sparsely populated Aristazabal Island, herring roe access (based on Spawn Habitat Index, SHI) is near or above the average SHI across a historic baseline (S3 Fig). Commercial marine fisheries access (via First Nations commercial, Northern Native Fishing Corporation, and communal-commercial licenses) increased steadily in all regions (S3 Fig). Shellfish access was typically the lowest-scoring component across regions and years, particularly for Strait of Georgia and West Vancouver Island regions, and predominantly due to biotoxin-related closures (S3 Fig). It should be noted that some of these closures may be due to limited testing capacity [40], rather than reflective of actual toxin presence, though even a precautionary closure still restricts access to the resource.

Our Livelihoods results reflect an economic disparity between BC's First Nations and non-First Nations communities, with First Nations livelihoods scoring lower than non-First Nations (S1 Fig), as measured by both median inflation-adjusted income and employment rate. Between 2006 and 2016, income is generally stable or increasing for all regions for both First Nations and non-First Nations communities, while unemployment is generally stable or decreasing in all regions. However, while overall livelihoods seem to be improving, the gap between First Nations and non-First Nations, reflected in the difference between First Nations Livelihoods and Non-First Nations Livelihoods scores, is generally persistent over time. As noted previously, non-monetary benefits to livelihoods are often quite important across British Columbia [13] but are not accounted for within this goal due to lack of available quantitative data.

With the exception of the North Coast, Tourism & Recreation scores are relatively stable, reflecting modest changes over time in attendance at both coastal parks and coastal visitor centers. North Coast visitor center visits dropped by nearly half between 2009 and 2010, followed by a similar drop in park visits between 2010 and 2013. The Central Coast and Aristazabal Island regions are not represented in these results due to lack of data on visitor center and park attendance. Note that data on park and visitor center visits were only available from 2007 onward, and the no-net-loss model therefore scores all regions at 100 for this goal in 2007.

Sense of Place scores generally show a dramatic rise over the course of the study period, driven entirely by rapid improvements in Lasting Special Places scores. A rapid pace of designation of additional protected marine and coastal area is evident in all regions but Strait of Georgia. Most notable were the sizeable expansion of terrestrial and marine protection in a network of heritage sites/conservancies around Haida Gwaii in 2008 and the designation of Gwaii Haanas National Marine Conservation Area Reserve in 2010. Furthermore, all northern

OHI regions have provided regional plans outlining special management zones important for marine spatial planning in the regions [21–24]. After the study period, a new federal MPA, Hecate Strait and Queen Charlotte Sound Glass Sponge Reefs Marine Protected Area, was designated in 2017 [41], and another large candidate MPA off the continental shelf is being prioritized for designation by 2020 to protect offshore seamounts and hydrothermal vents [42]. Unlike Lasting Special Places scores, Iconic Species scores are nearly static, since IUCN and COSEWIC assessments are updated infrequently; of all 42 identified iconic species, no change in conservation status was reported within the study period. All observed changes in Iconic Species score are related to pressures and resilience. The average of static Iconic Species scores and rapidly increasing Lasting Special Places scores results in strong gains for the Sense of Place goal for nearly every region over the study period.

Biodiversity subgoals (Habitats, Species) both score high for all regions and years. Most species are found throughout the entire BC region, resulting in nearly identical Species subgoal scores among regions. Habitat scores, as for the Habitat Services subgoals, reflect the relatively intact coastal environment of BC since 1990. A BC-wide 2012 trawl habitat agreement among industry, DFO, and conservation organizations has dramatically curtailed trawling impacts on vulnerable and rare habitats [43].

Clean Waters scores were generally high and stable, reflecting low pressures from pathogen and nutrient components moderated by rather high pressures from chemicals (e.g. shipping pollution and land-based runoff and pesticides) and marine debris (S4 Fig). Pressure from marine debris drives down scores in all regions, though this component is based on a static model of small plastics presence using 2014 data [44] so does not reflect changes over time. In the Strait of Georgia, higher development and population density further drive down scores due to high chemical pressures. Annual variation is primarily driven by changes in nutrient pressure; other components are generally stable and/or based on static datasets. The relatively low Clean Waters score for the sparsely populated Aristazabal Island region reflects surprisingly high localized marine debris pressure based on the plastic debris model [44].

One of the strengths of the OHI framework is that it allows for direct comparison among disparate goals and regions within an assessment. Comparisons among different assessments can also be made, but with caution because of the different goals assessed, data used, reference points established, and scale of assessment, among other differences. So although Canada's Pacific EEZ has also been included in global assessments [1,4,5] and a Canada-wide OHI [10] that assesses Canada's entire EEZ as a single entity, and ecological and socio-economic similarities to a US West Coast OHI [11] could offer useful insights, we do not address those comparisons here as they are beyond the scope of this paper.

The Ocean Health Index process provides insight into the state of the coupled marine social-ecological system beyond just the scores. The process of conducting an OHI assessment for British Columbia required explicit and transparent incorporation of the wide range of ocean management goals from the perspective of management agencies, scientists, conservation organizations, and all those who realize benefits from access to a healthy ocean. Discussions towards that end often revealed different values and philosophies that required reconciliation—for example, should the Food Provision goal include all seafood produced whether for local consumption or export (i.e., total production) or focus on food security within BC (i.e., local consumption only—but then what about imports)? In the Wild Capture Fisheries sub-goal, should underfishing (i.e., fishing a healthy stock below its maximum sustainable harvest) be penalized for not maximizing the sustainable harvest of available food, or should allowances be made for underfishing due to economic, cultural, or conservation reasons? Should coastal forests be considered "marine-associated habitats" for the sake of coastal protection, carbon storage, or biodiversity? And frequently, are there data available to

accurately communicate progress toward a particular value that meet spatial and temporal needs? Often, decisions involved a tradeoff between data availability against optimally representing all potential views and values. While we acknowledge the inherently normative nature of such decisions, we have presented our decisions explicitly and transparently within our goal models and data preparation (see SI Methods), as a starting point for future conversations.

Perhaps the most difficult, and yet the most important, step in conducting OHI assessments is establishing reference points for goal models [7,25]. Some goals lend themselves well to reference points based on scientific understanding, e.g., $B_{MSY}$ and $F_{MSY}$ as targets for sustainable management of wild-capture fisheries, based on a globally recognized, long established fisheries management framework [45]. Other goals rely on comparisons to historical baselines, e.g., herring spawn habitat index in the First Nations Resource Access Opportunity goal or coastal forests and salt marsh in the Habitat Services goal. However, historical baselines require decisions as to what point in the past the baseline should represent (pre-colonization, pre-industrial, post-war, etc.), and are often constrained by inclusive consensus on the appropriate time period, as well as data availability. In yet other cases, explicit management targets provide guidance, e.g., harvest and escapement targets in Pacific salmon management. However, for many goals, an ideal reference point is a normative question: what is the "right" amount of unemployment or tourism? Is it fair to compare employment rates and incomes between the urbanized Strait of Georgia and sparsely populated Central Coast? In these cases, value judgments will differ, and different choices will result in different scores.

Giving explicit voice to ideals of ocean health, and to reference points against which these ideals can be measured, highlights in many cases the gaps and lack of critically important data (see SI methods, S13 and S14 Tables). To ensure full transparency and ease of repeatability, OHI assessments use free and openly available datasets wherever possible [12]. In many cases, "ideal" data for a particular indicator do not exist or were not openly available, requiring the use of alternative proxies. In other cases, data were not available for the entire time series of our assessment (Fig 2) or for the entire study region, resulting in tradeoffs between spatial and temporal extent and resolution within this assessment. While such compromises are inevitable, they highlight the value of a comprehensive assessment as a form of gap analysis to prioritize improvements in monitoring and data collection to better inform our understanding of ocean health.

Just as individuals may prioritize ocean health benefits differently, they may also experience the provision of those benefits at different scales; e.g., a fisher may be concerned with the conditions at a favorite fishing spot, but the harvest will also be dependent upon the conditions throughout the larger region. By aggregating data up to the scale of the defined OHIBC regions, goal scores may reflect the general state of ocean resources but fail to capture heterogeneity in ocean health at the community or individual scale. Tension between narrative and lived experience on the one hand and analysis at the scale of data availability on the other may indicate an opportunity to refine goal models, redefine reference points, conduct assessments at even finer scales, or generate relevant data and insights through methods such as in-depth interviews or deliberative mapping [46]. Another tradeoff exists between the scale of assessment and the scale at which governance occurs. While a community-level application of OHI (assuming appropriate data availability) may convey a more locally relevant impression of ocean health, community-scale results are less likely to drive province-level or federal-level improvements in ocean management.

### Assessing management within the OHI framework

The Ocean Health Index framework provides a unique and powerful tool for comprehensively assessing a baseline status of coupled social-ecological systems in coastal regions, accounting

for the relationship between humans and the marine environment. Repeating this assessment over time allows one to track changes in scores and, ideally, see the consequences of management actions reflected in those scores. Indeed, underlying most indicator efforts and general thinking around sustainable ocean management is an assumption that pressures and resilience metrics should have measurable consequences for the things we care about with respect to ocean health. The OHI framework is unique in that scores explicitly account for these predicted impacts in a way that these assumptions can be tested, given appropriate data as inputs.

Supporting expectations that effective management can reduce pressures on ecological systems, we found that resilience conferred by regulatory measures (i.e., marine protected areas, aquaculture regulations, and fishing regulations) and ecological integrity corresponded with significant reductions in future pressures for the Wild-Capture Fisheries subgoal and the Habitat Services subgoals (i.e. Coastal Protection and Carbon Storage) (Table 2). Additionally, social resilience significantly correlates with a decrease in pressures for Habitat Services. Allowing for longer lag times (i.e., greater than one year) between resilience and observed effect on pressure (S6 Table), we see additional evidence for effectiveness of management on Wild-Capture Fisheries, Aquaculture, Wild-Capture Salmon, and Biodiversity goals, and benefits of social resilience on Tourism & Recreation and Lasting Special Places. These results support the idea that the OHI framework has the potential to be sensitive to the impacts of management action on the social-ecological system. However, significant effects of reduced ecological pressures on goal status were not observed.

We did not see the expected relationships between resilience, pressures, and status for most goals. Three possible explanations for this result are: 1) OHI is not constructed to be sufficiently sensitive to change; 2) the changes in the system (natural or management driven) were not as impactful as expected or fast enough to be detectable; or 3) indicator data were insufficient (i.e., data limited or poor proxies) to observe correlation with real changes in ocean health. We do not believe the first reason to be true, as sensitivity analyses show OHI scores to change when underlying data change [4]. The second reason is certainly possible. We know that fisheries management can demonstrably improve sustainability of stocks [47] and ecosystem health [48], that MPAs can benefit biodiversity [49,50], and that aquaculture regulations can reduce ecosystem impacts of farmed seafood provision [51]. However, measurable responses in ocean health to such management actions can be slow, difficult to detect at large assessment scales, and confounded by other pressures and impacts beyond the boundaries of our coastal social-ecological system. Additionally, the time scale of interactions among management efforts, social resilience, and ecological and social pressures is almost certainly far more complicated than our simple single-lag (i.e., $\lambda$) response model allows.

As such, the lack of demonstrable response to management action for most goals is likely due at least in part to insufficient data available to quantify resilience, pressures, and/or status at the temporal and spatial scale of our assessment. Even in the context of a marine-dependent social-ecological system in a wealthy modern economy, we encountered challenges in accessing environmental, social, and economic data at spatial, sectoral, and temporal resolutions relevant to conservation policy makers (see SI Methods, S13 and S14 Tables). For example, several habitat layers are based on land use change data taken at ten-year intervals, and thus cannot be responsive to annual-scale changes in pressure. Similarly, income and employment data from the Canada census are collected every five years. Data to quantify enforcement and compliance with conservation-oriented management and regulation were particularly challenging to find, and even when data were available, they generally lacked reference points grounded in policy goals. For example, to quantify monitoring and enforcement of fisheries regulations, we used data on at-sea monitoring (for groundfish fisheries) and the number of fisheries officers employed each year, relative to the size of the commercial fishing fleet (for

other fisheries). BC's Integrated Groundfish Program mandates (and achieves) a 100% target for at-sea and dockside monitoring [43,52]. However, for fisheries not covered under this program, targets to define the number of officers required to ensure adequate enforcement of regulations across fisheries were not available for our analysis. Additional information such as spatial distribution of enforcement effort, budgetary allocation toward resources, or number of violations cited, would further improve our understanding of enforcement and compliance for non-groundfish fisheries.

Several broad opportunities exist to improve the quality and availability of social and ecological data to further future ocean health assessments in British Columbia. Numerous community-level efforts to monitor and manage local marine resources, such as the Coastal Guardian Watchmen initiative of Coastal First Nations [53] or the Great Canadian Shoreline Cleanup [54] could be coordinated coast-wide to systematize collection and communication of data so that it is comparable from context to context and made publicly available. Required data are often held on internal servers distributed across many institutions and departments in non-standard formats and/or retained by scientists reluctant to give up hard-won data [39]. While some datasets are subject to privacy protections, aggregation to appropriate spatial, temporal, or population scales would ensure the privacy and security of sensitive information. Government mandates for public reporting of taxpayer-funded data could be facilitated by enlisting the help of open-science researchers and institutions to clean, process, and publish data in publicly discoverable and readily accessible databases. Canada's Open Canada site (https://open.canada.ca/en/open-data) demonstrates considerable progress toward transparency and accountability goals of the 2018–2020 National Action Plan on Open Government [55].

As a composite index, OHIBC, like all OHI assessments, highlights connections among marine management goals, in particular the cross-cutting nature of anthropogenic pressures and resilience measures that affect delivery of multiple ocean-derived benefits simultaneously. Investment in management actions, high quality data collection, and long-term monitoring of a particular aspect of ocean health are likely to provide direct and indirect value to other facets of the marine social-ecological system. Accounting for this additional value could have important implications for the prioritization and distribution of resources within and across agencies and organizations. For example, MPAs that protect local biodiversity may support the presence of iconic species, which draw tourists and thus support healthy local economies, leading to improved social resilience, which may in turn reduce pressures on a broad array of other ocean-derived benefits.

## Conclusion

This longitudinal application of the OHI framework to Canada's Pacific waters succinctly communicates a wealth of information about British Columbia's ocean health for a range of potential audiences, fostering a holistic and interconnected view of the relationship of people with their coastal environment. While British Columbia enjoys a reputation for environmental stewardship, OHIBC highlights challenges and opportunities for improvement, particularly in terms of food provision and First Nations access to marine resources and economic opportunities. Our analysis reveals that OHIBC may be sensitive enough in some cases to attribute improvements in ocean health to regulatory action, particularly as better data become available. Challenges in data availability highlight areas in which coordinated efforts among federal and province agencies, First Nations, and conservation organizations could further improve data collection and dissemination to better serve marine resource management. Importantly, this assessment is not intended to be a final, static product. We hope that local scientists, policy

makers, and stakeholders will build upon and improve upon our work to ensure the long-term health of British Columbia's coastal social-ecological system. The flexibility of the OHI framework to adapt to improved understandings in the form of improved goal models or finer scale data sets, combined with the open science principles of transparency and reproducibility, make OHIBC an important stepping stone for continued monitoring, management, and assessment of British Columbia's ocean health.

## Supporting information

**S1 File.**
(PDF)

## Acknowledgments

We thank the National Center for Ecological Analysis and Synthesis (NCEAS) for computational support; Joy Wade for data preparation and management; and Andrew Day and the Vancouver Aquarium Marine Science Centre for insights and feedback. We appreciate the support of Fisheries and Oceans Canada (DFO), in particular DFO salmon manager Jeff Grout and salmon science experts Mary Thiess, Anne-Marie Huang, Katarina Wor, and Sue Grant. We appreciate thorough reviews from Sarah Dudas, Jennifer Boldt, and Sebastian Villasante, whose comments helped improve and clarify this manuscript.

## Author Contributions

**Conceptualization:** Casey C. O'Hara, Courtney Scarborough, Jamie C. Afflerbach, Melanie Frazier, Julia S. Stewart Lowndes, Benjamin S. Halpern.

**Data curation:** Casey C. O'Hara, Courtney Scarborough, Karen L. Hunter, Jamie C. Afflerbach, Karin Bodtker, Melanie Frazier, R. Ian Perry.

**Formal analysis:** Casey C. O'Hara, Karen L. Hunter, Jamie C. Afflerbach, Melanie Frazier, Benjamin S. Halpern.

**Funding acquisition:** Benjamin S. Halpern.

**Investigation:** Casey C. O'Hara, Courtney Scarborough, Karen L. Hunter, Jamie C. Afflerbach, Melanie Frazier, Benjamin S. Halpern.

**Methodology:** Casey C. O'Hara, Courtney Scarborough, Karen L. Hunter, Jamie C. Afflerbach, Karin Bodtker, Melanie Frazier, Julia S. Stewart Lowndes, R. Ian Perry.

**Project administration:** Casey C. O'Hara, Courtney Scarborough, Benjamin S. Halpern.

**Software:** Casey C. O'Hara, Jamie C. Afflerbach, Melanie Frazier, Julia S. Stewart Lowndes.

**Supervision:** R. Ian Perry, Benjamin S. Halpern.

**Validation:** Casey C. O'Hara, Jamie C. Afflerbach, Melanie Frazier.

**Visualization:** Casey C. O'Hara, Courtney Scarborough, Jamie C. Afflerbach, Melanie Frazier, Julia S. Stewart Lowndes, Benjamin S. Halpern.

**Writing – original draft:** Casey C. O'Hara, Courtney Scarborough, Benjamin S. Halpern.

**Writing – review & editing:** Casey C. O'Hara, Courtney Scarborough, Karen L. Hunter, Jamie C. Afflerbach, Karin Bodtker, Melanie Frazier, Julia S. Stewart Lowndes, R. Ian Perry, Benjamin S. Halpern.

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
