## [Decision Letter · Decision Letter 0]

8 Oct 2019

PONE-D-19-24580

Changes in ocean health in British Columbia from 2001 to 2016

PLOS ONE

Dear Mr. O'Hara,

Thank you for submitting your manuscript to PLOS ONE. After careful consideration, we feel that it has merit but does not fully meet PLOS ONE’s publication criteria as it currently stands. Therefore, we invite you to submit a revised version of the manuscript that addresses the points raised during the review process.

In your revision it would be interesting if you could include some information regarding the point raised by reviewer #1 on management in terms of stakeholder's participation including local government administrators in the OHI process for this region.

We would appreciate receiving your revised manuscript by Nov 22 2019 11:59PM. To enhance the reproducibility of your results, we recommend that if applicable you deposit your laboratory protocols in protocols.io, where a protocol can be assigned its own identifier (DOI) such that it can be cited independently in the future. For instructions see: http://journals.plos.org/plosone/s/submission-guidelines#loc-laboratory-protocols

We look forward to receiving your revised manuscript.

Kind regards,

Andrea Belgrano, Ph.D.

Academic Editor

PLOS ONE

Journal Requirements:

1. We note that you have stated that you will provide repository information for your data at acceptance. Should your manuscript be accepted for publication, we will hold it until you provide the relevant accession numbers or DOIs necessary to access your data. If you wish to make changes to your Data Availability statement, please describe these changes in your cover letter and we will update your Data Availability statement to reflect the information you provide.

Reviewers' comments:

Reviewer's Responses to Questions

**Comments to the Author**

1. Is the manuscript technically sound, and do the data support the conclusions?

Reviewer #1: Yes

2. Has the statistical analysis been performed appropriately and rigorously? 

Reviewer #1: Yes

3. Have the authors made all data underlying the findings in their manuscript fully available?

Reviewer #1: Yes

4. Is the manuscript presented in an intelligible fashion and written in standard English?

Reviewer #1: Yes

5. Review Comments to the Author

Reviewer #1: I really enjoyed reading this interesting piece of work. I found it appropriate and necessary, specially to address the regional scale of the OHI.

However, I think that the paper would benefit from explaining in more detail the incomplete datasets. For example, in lines 149-151 authors say "In such cases, datasets were chosen based on careful consideration of the tradeoffs among spatial resolution, temporal resolution, and how well the data represented the needs of the assessment." Could you provide a specific example of how you did it?

Again, in lines 169-172 the authors state that "To address incomplete data sets, we applied two gapfilling procedures. For periodic data, we estimated intervening years using a linear interpolation between available data years. For truncated data, we typically expanded the time series using last observation carried forward and/or first observation carried back methods". How you did it? By using expert consultation? Did other stakeholders validated the results? How?

Lines 525-528: the lack of cultural availability of data is an important gap, but at the same time, an excellent window of opportunity to collect data through, e.g., in-depth interviews or deliberative mapping.

Management: did the authors discuss the process of the OHI with regional/local representatives of the administration? Would be really strong to include this in more detail into the discussion, because it will help policy makers to encourage them to use the OHI.

Looking forward to the next version of the manuscript.

6. PLOS authors have the option to publish the peer review history of their article (what does this mean?). If published, this will include your full peer review and any attached files.

Reviewer #1: Yes: Sebastian Villasante

---

## [Author Response · Author response to Decision Letter 0]

18 Nov 2019

See "response to reviewers"; text pasted below:

Thank you for the thoughtful feedback on our manuscript, “Changes in ocean health in British Columbia

from 2001 to 2016.” We have addressed all the reviewer’s concerns and suggestions, and feel that the

resulting manuscript is now much stronger. Below, we outline the changes we have made in response to the

reviewer’s suggestion.

Reviewer #1: I really enjoyed reading this interesting piece of work. I found it appropriate and necessary,

specially to address the regional scale of the OHI.

However, I think that the paper would benefit from explaining in more detail the incomplete datasets. For

example, in lines 149-151 authors say “In such cases, datasets were chosen based on careful consideration of

the tradeoffs among spatial resolution, temporal resolution, and how well the data represented the needs of the

assessment.” Could you provide a specific example of how you did it?

We have now included details on this process in the Supporting Information Methods, in its own section

titled “Data selection criteria” starting on line 559. We rank spatial resolution and extent, temporal resolution

and extent (and baseline where appropriate), and thematic “fit” and “resolution” of each dataset to its

particular status, pressure, or resilience calculation. Each dimension is scored on a 0.0/0.5/1.0 scale based on

criteria described in the supporting methods. In S13 Table, the relevant scores are averaged for each dataset

as a heuristic for selection and comparison. As an example of using this method as data selection criteria, we

compare two datasets included in the OHIBC assessment against two that we considered, but decided

against.

In S14 Table, we aggregate dataset scores to calculate scores for each goal status, pressure, and resilience

layers.

Within the manuscript itself, we have included notes referring to these SI Methods in several locations:

• Lines 150-153: “In such cases, datasets were chosen based on careful consideration of the tradeoffs

among spatial extent and resolution, temporal extent and resolution, and how well the data represented

the needs of the assessment (see SI methods, S13 and S14 Tables).”

• Lines 610-612: “Giving explicit voice to ideals of ocean health, and to reference points against which

these ideals can be measured, highlights in many cases the gaps and lack of critically important data (see

SI methods, S13 and S14 Tables).”

• Lines 673-675: “…we encountered challenges in accessing environmental, social, and economic data at

spatial, sectoral, and temporal resolutions relevant to conservation policy makers (see SI Methods, S13

and S14 Tables).”

Note also changes in lines 248-252 to include both spatial and temporal extent and resolution (original text just

mentioned resolution.)

Again, in lines 169-172 the authors state that “To address incomplete data sets, we applied two gapfilling

procedures. For periodic data, we estimated intervening years using a linear interpolation between available

data years. For truncated data, we typically expanded the time series using last observation carried forward

and/or first observation carried back methods”. How you did it? By using expert consultation? Did other

stakeholders validated the results? How?

We chose linear interpolation and Last Observation Carried Forward/Next Observation Carried Back

(LOCF/NOCB) as simple, transparent gapfilling procedures that are commonly used including in global OHI

assessments. To make our methods more transparent, we have explicitly noted in the SI Methods where

linear interpolation and/or LOCF/NOCB extrapolation have been used for each goal. In the line-numbered

version of SI (attached for convenience), see lines 83, 103, 141, 161, 172, 195, 204, 209, 221, 226, 267, 281,

295, 325, 349, 368, 375, 388, 397. The text of these additions is omitted here for brevity. Note also minor

changes to the manuscript text in lines 272-275: “To address incomplete data sets, we applied two gapfilling

procedures chosen for simplicity and transparency. For periodic data, we estimated intervening years using a linear

interpolation between available data years. For truncated data, we typically expanded the time series using last

observation carried forward and/or next observation carried back extrapolation methods.”

Lines 525-528: the lack of cultural availability of data is an important gap, but at the same time, an

excellent window of opportunity to collect data through, e.g., in-depth interviews or deliberative mapping.

This is an excellent point - we have added a reference in line 626 to highlight this opportunity: “…may

indicate an opportunity to refine goal models, redefine reference points, conduct assessments at even finer

scales, or generate relevant data and insights through methods such as in-depth interviews or deliberative

mapping.”

Management: did the authors discuss the process of the OHI with regional/local representatives of the

administration? Would be really strong to include this in more detail into the discussion, because it will help

policy makers to encourage them to use the OHI.

This is an excellent suggestion, but we do not wish to overstate the role of administrators in the process. In

the initial phases of this assessment, we discussed the OHI process with representatives of First Nations and

the Province of British Columbia via MaPP, engaging them to help define the goals and align them with local

and regional needs and priorities. Beyond this planning phase, our engagement with administration has

primarily been in a scientific advisory role through DFO scientists (including coauthors Karen Hunter and R.

Ian Perry) rather than policymakers. Because our funding source is also involved in MaPP, we wished to

maintain a degree of independence between the planning efforts of MaPP and the resulting calculations of

OHIBC, to avoid the appearance of a conflict of interest.

Thank you for your time and effort in reviewing our manuscript. We believe our revisions have addressed all

comments thoroughly.

---

## [Editor Report · Decision Letter 1]

20 Dec 2019

Changes in ocean health in British Columbia from 2001 to 2016

PONE-D-19-24580R1

Dear Dr. O'Hara,

We are pleased to inform you that your manuscript has been judged scientifically suitable for publication and will be formally accepted for publication once it complies with all outstanding technical requirements.

With kind regards,

Andrea Belgrano, Ph.D.

Academic Editor

PLOS ONE

Additional Editor Comments (optional):

Thank you for addressing ALL the comments and suggestions in your revised manuscript

---

## [Editor Report · Acceptance letter]

6 Jan 2020

PONE-D-19-24580R1 

Changes in ocean health in British Columbia from 2001 to 2016 

Dear Dr. O'Hara:

I am pleased to inform you that your manuscript has been deemed suitable for publication in PLOS ONE. Congratulations! Your manuscript is now with our production department. 

With kind regards,

on behalf of

Dr. Andrea Belgrano 

Academic Editor

PLOS ONE